# Double-Bayesian Learning

## Abstract

Contemporary machine learning methods will try to approach the Bayes error, as
it is the lowest possible error any model can achieve. This paper postulates that
any decision is composed of not one but two Bayesian decisions and that decision-
making is, therefore, a double-Bayesian process. The paper shows how this duality
implies intrinsic uncertainty in decisions and how it incorporates explainability.
The proposed approach understands that Bayesian learning is tantamount to finding
a base for a logarithmic function measuring uncertainty, with solutions being fixed
points. Furthermore, following this approach, the golden ratio describes possible
solutions satisfying Bayes' theorem. The double-Bayesian framework suggests
using a learning rate and momentum weight with values similar to those used in
the literature to train neural networks with stochastic gradient descent.

## 1 Introduction

Despite the progress in machine learning, several problems stand out for which convincing solutions
have yet to be found. With massive training sets, enormously sized networks, and immense computing
power, training machine learning models has become a brute force approach, arguably more concerned
with memorization than generalization. However, quoting from a post by Y. LeCun (Nov. 23, 2023),
we know that

*Animals and humans get very smart very quickly with vastly smaller amounts of training data than
current AI systems. Current large language models (LLMs) are trained on text data that would take
20,000 years for a human to read. And still, they haven't learned that if A is the same as B, then B
is the same as A. Humans get a lot smarter than that with comparatively little training data. Even
corvids, parrots, dogs, and octopuses get smarter than that very, very quickly, with only 2 billion
neurons and a few trillion "parameters."*

This raises the question of whether modern training techniques and principles are actually biologically
implemented in the human brain and, if not, what alternative methods could save resources. More
efficient methods would be better at generalizing with smaller amounts of training data, which almost
certainly would also improve the explainability and interpretability of neural networks.

This paper investigates what it takes for a classifier to be optimal. The starting point is Bayes' theorem,
which is the foundation of the Bayes classifier. The Bayes classifier is considered optimal because
it minimizes the Bayes risk, meaning it has the smallest probability of misclassification among all
classifiers. However, applying the Bayes classifier directly is often impossible because of the difficulty
in computing the posterior probabilities. For this reason, most classifiers are trying to approximate
the Bayes classifier, like the naïve Bayes classifier, for instance. The information-theoretical analysis
presented in this paper splits the decision of a Bayes classifier into two decisions, each following
Bayes' theorem, where one decision can serve as an explanation or verification of the other. Each of
the two decision processes faces intrinsic uncertainty, as its decision depends on the output of the
other process. The paper will investigate the theoretical ramifications of this approach. As a practical

result, it will discuss the consequences for two hyperparameters of stochastic gradient descent used in the training process of a neural network: learning rate and momentum weight.

The structure of the paper is as follows: After this introduction, Section 2 motivates one of the main ideas, namely that learning to make a decision involves solving two sub-problems and, thus, two decisions. Section 3 discusses Bayes' theorem, which is central to statistical decision-making and is the starting point of the theoretical approach outlined in the following. Section 4 then introduces the double-Bayesian model as the key concept of the paper. The next section, Section 5, shows how to represent possible solutions of the double-Bayesian decision model. Section 6 discusses the golden ratio, including its functional equations and how it defines a solution to the double-Bayesian model. Then, Section 7 discusses the theoretical implications for training double-Bayesian networks with stochastic gradient descent. Finally, Section 8 summarizes the key concepts, followed by a conclusion.

## 2  Dual decisions

Suppose a sender transmits the image on the left-hand side of Figure 1 to a receiver. This image

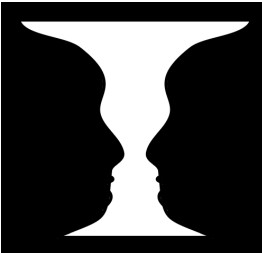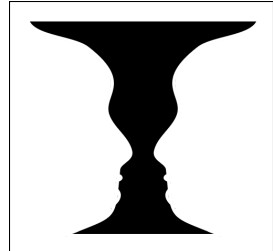

Figure 1: An image of Rubin's vase (left) and its inverted counterpart (right) - (Rubin, 1915)

depicts *Rubin's vase* by the Danish psychologist Edgar Rubin (Rubin, 1915), which shows a vase or two faces looking at each other, depending on the receiver's perception. The receiver then faces an unsolvable conundrum: 1) If the receiver thinks the image represents a vase, the receiver cannot be certain that the vase is indeed the intended message the sender wanted to convey. Maybe the sender wanted to send the faces. 2) If the receiver is expecting a picture of a vase (or faces) and thus knows the intended message, there is no certainty that an image of a vase has been transmitted. After all, the image could show faces. Therefore, two decisions are involved in making the final interpretation of the image: 1) a decision about the perception of the image (vase or faces), and 2) a decision about whether the perceived image coincides with the intended message, meaning the image transmitted. Both decisions together are fraught with intrinsic uncertainty because deciding the ultimate interpretation of Rubin's vase, a vase or faces, is impossible. Therefore, neither the sender nor the receiver can make both decisions without uncertainty. Instead, the knowledge is distributed. The sender knows the intended message (a vase or faces) but not the receiver's perceived image. On the other hand, the receiver knows the perceived image (a vase or faces) but not the intended message. Therefore, the sender and the receiver must collaborate to get the true interpretation across their communication channel.

Let the sender and receiver perceive Rubin's vase differently, with contrary opinions about the foreground and background color (black or white), where the foreground represents the perceived image, either a vase or faces. Furthermore, let the sender and the receiver both be able to send an image of Rubin's vase to each other so that both become senders and receivers alike and can share their knowledge about the perceived image and intended message. The image that the sender perceives is then the inverted image that the sender perceives. The goal is to collaborate so that the perceived image (foreground) equals the intended message on both ends.

A sender can either send the image of Rubin's vase on the left-hand side of Figure 1 or send the image with colors inverted, as shown on the right-hand side of Figure 1, depending on the perceived image or intended message, respectively. On the other end, the receiver has two options: 1) accept the received image if it is identical to the image expected, or 2) tell the sender to invert the image if it is different. After this feedback, the image on the receiver end will be the same as the image on the

sender side. By making the images on both sides the same, the receiver has completed half of the decision process without making a mistake and has thus behaved optimally. The receiver has ensured that both sides see the same image. It is now up to the sender to make the final, second decision about what image needs to be inverted to arrive at the final interpretation, either the image of the sender or the image of the receiver. Thus, the first process tries to make the images identical, whereas the second process tries to make the images different on both ends to reflect the different perceptions of the sender and receiver.

Although described as a sequential process, the two dual decision processes leading to the final interpretation are running in parallel. The sender is also a receiver, and the receiver is also a sender. One of them conveys the correct foreground information (black or white), while the other conveys the message. Note that neither the sender nor the receiver will ever see the true interpretation of the image. The receiver in the example above will never know whether the received image needs to be inverted after making the images identical because this would mean the receiver knows the true interpretation of the image, which is not possible according to the uncertainty principle described above. A similar statement can be made for the sender. The sender and the receiver can be considered dual and complementary forces because of their different interpretations of foreground and background. They make two binary decisions, deciding on the correct foreground color (black or white) and on the message (a vase or faces). They decide whether Rubin's vase should be interpreted as a white vase, a black vase, white faces, or black faces.

## 3 Bayes theorem

Bayes' theorem is a fundamental law in probability theory that describes the probability of an event given prior knowledge. The theorem is of central importance in machine learning, where it guides the training of machines for decision-making, such as in Bayesian inference or naïve Bayes classification. For two events $A$ and $B$, with prior probabilities $P(A)$ and $P(B)$, and $P(B) \neq 0$, Bayes' theorem states the following:

$$P(A|B) = \frac{P(A) \cdot P(B|A)}{P(B)}, \tag{1}$$

where $P(A|B)$ and $P(B|A)$ are the conditional or posterior probabilities. Thus, $P(A|B)$ is the probability of event $A$ occurring when $B$ is true, and analogously, $P(B|A)$ is the probability of $B$ given that $A$ is true.

For a machine learning application, $A$ would be the class of an observed input pattern $B$. The probability $P(A)$ is then the prior probability of class $A$, and $P(B)$ is the prior probability of seeing pattern $B$. Consequently, $P(A|B)$ is the posterior probability of class $A$ when seeing pattern $B$, and $P(B|A)$ is the posterior probability of $B$ within $A$. According to Bayes' theorem, three probabilities are needed to compute the probability $P(A|B)$ that class $A$ is observed when seeing pattern $B$: $P(A)$, $P(B)$, and $P(B|A)$. However, several obstacles prevent Bayes' theorem from being applied in this way. No particular method can help determine the prior probabilities, which are often unknown. Furthermore, the posterior probability is often not readily available and is approximated by making assumptions about the distribution of $B$ given $A$, for example, assuming a normal distribution.

To cope with these limitations, the next section describes decision-making as a dual process based on Bayes' theorem, with uncertainty intrinsically involved.

## 4 Double-Bayesian framework

The Bayes Theorem is typically stated as in Eq. 1. However, restating the theorem in the following equivalent form highlights the two decision processes for the two subproblems involved, as motivated in Section 2:

$$\frac{P(A|B)}{P(B|A)} = \frac{P(A)}{P(B)} \tag{2}$$

The left-hand side of Eq. 2 features a fraction of the posterior probabilities, whereas the right-hand side shows the prior probabilities. Following the motivation in Section 2, the posterior probabilities, $P(A|B)$ and $P(B|A)$, can be understood as the probability that $A$ or $B$ is the intended message, respectively. Then, the prior probabilities, $P(A)$ and $P(B)$, would express the probabilities that $A$ or $B$ is in the foreground.

With only one equation for four parameters, Eq. 2 is underdetermined. However, it is fair to assume that $1 - (P(A|B) = P(B|A)$ and $1 - P(A) = P(B))$, which leaves one equation with one parameter on each side. This is possible because either $A$ or $B$ can be the message or foreground, not both of them at the same time, following again the reasoning in Section 2. Therefore, the intrinsic uncertainty in Bayes' theorem can be described as follows: if the true foreground is known, then whether the message needs to be swapped is unknown; on the other hand, if the message is known, then whether the foreground needs to be swapped is unknown. The fractions on both sides of Eq. 2 are thus "cognitively entangled."

The two remaining unknown parameters can be computed using two separate processes, each adding a constraint to handle the uncertainty. To illustrate this, Eq. 3 restates Bayes' theorem in yet another way:

$$1 = \frac{P(A)}{P(B)} \cdot \frac{P(B|A)}{P(A|B)} \tag{3}$$

Assuming that $P(B) = P(B|A)$, Eq. 3 simplifies to $P(A|B) = P(A)$. This assumption of $B$ being independent of $A$ is fair because, according to the motivation in Section 2, the decisions about the message and the foreground are independent of each other. Under this assumption, only one unknown remains, either $P(A|B)$ or $P(A)$, which follows directly from either $P(A)$ or $P(A|B)$, depending on which is input and which is output.

A similar, symmetric statement can be made when using the reciprocals on both sides of Eq. 2, which leads to the following equation:

$$1 = \frac{P(B)}{P(A)} \cdot \frac{P(A|B)}{P(B|A)} \tag{4}$$

Here, assuming that $A$ is independent of $B$ simplifies Eq. 4 to $P(B|A) = P(B)$.

Solving Eq. 2, Eq. 3, or Eq. 4 will be referred to as solving the outer Bayes equation. On the other hand, making both multiplicands on the right-hand side of Eq. 3 or Eq. 4 identical will be referred to as solving the inner Bayes equation, or simply solving the inner equation of Eq. 3 or Eq. 4. For Eq. 3, the inner Bayes equation thus states as follows:

$$\frac{P(A)}{P(B)} = \frac{P(B|A)}{P(A|B)} \tag{5}$$

Accordingly, the inner Bayes equation for Eq. 4 is obtained by using the reciprocals of the fractions on both sides of Eq. 5:

$$\frac{P(B)}{P(A)} = \frac{P(A|B)}{P(B|A)} \tag{6}$$

Consequently, the inner Bayes equations can derived by inverting a fraction on one side of Bayes' theorem, as stated in Eq. 2. The inner Bayes equations are thus "entangled" versions of Bayes' theorem.

The two independent decision processes motivated above are solving the inner and outer Bayes equations. To further formalize these processes, the following section will add a logarithmic expression to Eq. 3 and Eq. 4. Adding a logarithm offers several advantages: 1) using information theory to measure uncertainty; 2) using a reciprocal becomes equivalent to changing the sign of a logarithm; and 3) solving the equation in Bayes' theorem is reduced to finding a suitable base for a logarithm.

# 5 Fixpoint solutions

Using a logarithmic expression in Eq. 3 and Eq. 4 is possible when solutions become fixed points of a logarithmic function. To illustrate this, let $\log_b(x)$ be the logarithm for an input $x$ and a base $b$. By definition, the logarithm is the inverse function of taking the power. Therefore, the following equation holds:

$$x = \log_b(b^x) \tag{7}$$

For the base $b$ of a logarithm, any positive real number can be used so long as $b \neq 1$. A logarithm computed for base $b$ can be converted into a logarithm for base $b'$ as follows:

$$\log'_b(x) = \log_b(x) / \log_b(b') \tag{8}$$

168 Therefore, the simple term $\log$ is used for the logarithm in the following.

169 By applying the logarithm to probabilities, they become information. For the two dual processes
170 above, the information of one process will be its counterpart's information with a different sign. To
171 achieve this, the following identity is required:

$$\log(x) = x \tag{9}$$

172 The following lemma states that this requirement can be met for general input values.

173 *Lemma:* For every $x \in \mathbb{R}^+ \setminus \{1\}$, there exists a base $\lambda$ so that $\log_\lambda(x) = x$.

174 *Proof:* Let $b \in \mathbb{R}^+ \setminus \{1\}$ be an arbitrary basis for which $\log_b(x) = y$. Furthermore, let $k$ be
175 a multiplier so that $ky = x$. Then, $\log_\lambda(x) = x$ for $\lambda = b^{1/k}$. This follows from Eq.8, with
176 $\log_\lambda(x) = \log_b(x)/\log_b(\lambda) = \log_b(x)/\log_b(b^{1/k}) = \log_b(x) \cdot k = x$. $\square$

177 Note that the common logarithmic rules apply for a fixed $\lambda$. However, when requiring a $\lambda$ that always
178 satisfies $\log_\lambda(x) = x$, computations become ambiguous, as seen here: $-\log_\lambda(x) = -x \neq 1/x =$
179 $\log_\lambda(1/x)$. The base $\lambda$ should be understood as a dynamic parameter that a learning system can
180 modify over time so that $\log_\lambda(x)$ converges to the input $x$.

181 Using the $\log_\lambda$ expression of the above Lemma, the Bayes' equation in Eq. 3 can be written as
182 follows:

$$1 = \frac{P(A)}{P(B)} \cdot \log_\lambda \left( \frac{P(B|A)}{P(A|B)} \right) \tag{10}$$

183 Then, the following sequence of transformations can be derived from Eq. 10:

$$P(A|B) = \frac{P(A)}{P(B)} \cdot \log_\lambda \left( \frac{P(B|A)}{1} \right) \tag{11}$$

$$= \frac{1 - P(B)}{P(B)} \cdot \log_\lambda \left( P(B|A) \right) \tag{12}$$

$$= \left( 1 - P(B) \right) \cdot \log_\lambda \left( P(B)^2 \right) \tag{13}$$

$$= P(B) \cdot \log_\lambda \left( 1 - P(B)^2 \right) \tag{14}$$

$$= 2 \cdot P(B) \cdot \log_\lambda \left( \sqrt{1 - P(B)^2} \right) \tag{15}$$

$$= 2 \cdot \sin(\phi) \cdot \log_\lambda \left( \cos(\phi) \right), \tag{16}$$

184 where the last expression holds for an angle $\phi \in \left[ 0 ; \frac{\pi}{2} \right]$. The reasoning behind these transformations
185 is as follows:

186 The first step, Eq. 11, moves the posterior probability $P(A|B)$ back to the left-hand side of the
187 equation. The result is Bayes' theorem in its original form, as shown in Equation 1.

188 The next step, Eq. 12, replaces $P(A)$ with $1 - P(B)$, removing one degree of freedom as motivated
189 above.

190 In the same way, Eq. 13 reformulates Eq. 12, assuming that $P(B) = P(B|A)$ and that the two
191 multipliers on the right-hand side of the equation are equal to meet the inner Bayes equation.

192 Then, Eq. 14 rewrites the right-hand side of Eq. 13, transforming $1 - P(B) = P(B)^2$ into the
193 equivalent $P(B) = 1 - P(B)^2$, which must hold true to satisfy the inner Bayes equation.

194 Finally, Eq. 15 extracts a factor of two from the $\log_\lambda$ expression to get a radical input expression for
195 the logarithm, following the standard rules for logarithms. The new input term to the $\log_\lambda$ expression
196 in Eq. 15 allows visualizing all possible solutions to the outer and inner Bayes equations.

197 To illustrate this further, Eq. 16 rewrites Eq. 15 using trigonometric functions and the Pythagorean
198 relationship between $\sin$ and $\cos$: $\sin^2 \phi + \cos^2 \phi = 1$, and thus $\sin \phi = \pm\sqrt{1 - \cos^2 \phi}$ and
199 $\cos \phi = \pm\sqrt{1 - \sin^2 \phi}$. Solutions to the outer and inner Bayes equations then correspond to an
200 angle $\phi$ in Equation 16, depending on the base $\lambda$. Thus, solutions are points on the unit circle.
201 By changing the angle $\phi$ in Equation 16, all the possible solutions to the outer and inner Bayes

equations can be visualized. Following the reasoning above, the right-hand side of Eq. 16 represents the inner Bayes equation. Accordingly, after bringing the factor 2 on the other side of Eq. 16, the inner Bayes equation is satisfied when $\sin(\phi) = \cos(\phi)$, which is the case for $\phi = \pi/4$, with $\sin(\pi/4) = \cos(\pi/4) = 1/\sqrt{2}$.

For the dual process, the $\log_\lambda$ expression can be used in combination with the other term of the inner Bayes equation in Eq. 3, as shown here:

$$1 = \log_\lambda\left(\frac{P(A)}{P(B)}\right) \cdot \frac{P(B|A)}{P(A|B)} \tag{17}$$

Note that the $\log_\lambda$ expression has moved to the left compared to the right-hand side of Eq. 10. From this equation, the following sequence of transformations can be derived similar to the transformations above.

$$
\begin{aligned}
P(B) &= \log_\lambda\left(\frac{P(A)}{1}\right) \cdot \frac{P(B|A)}{P(A|B)} & (18)\\
&= \log_\lambda\left(P(A)\right) \cdot \frac{1 - P(A|B)}{P(A|B)} & (19)\\
&= \log_\lambda\left(P(A|B)^2\right) \cdot \left(1 - P(A|B)\right) & (20)\\
&= \log_\lambda\left(1 - P(A|B)^2\right) \cdot P(A|B) & (21)\\
&= 2 \cdot \log_\lambda\left(\sqrt{1 - P(A|B)^2}\right) \cdot P(A|B) & (22)\\
&= 2 \cdot \log_\lambda\left(\sin(\phi)\right) \cdot \cos(\phi) & (23)
\end{aligned}
$$

During this sequence, assumptions similar to the ones in Eq. 12 and Eq. 13 are made. In Eq. 19, $P(B|A)$ was replaced by $1 - P(A|B)$, and Eq. 20 assumes that $P(A) = P(A|B)$. Again, all transformations assume that both multiplicands on the right-hand side are equal to satisfy the inner Bayes equation.

The intrinsic uncertainty for the dual processes can again be seen in Eq. 16 and Eq. 23, where it manifests like this: if the base $\lambda$ is known, then the angle $\phi$ is unknown; and vice versa, if $\phi$ is known, then $\lambda$ is unknown. Each process contributes knowledge about $\lambda$ and $\phi$, which the other process does not know.

The process knowledge about $\lambda$ and $\phi$ does not need to be "all-or-nothing." The uncertainty ranges continuously between two extremes, and both dual processes can be somewhat knowledgeable about both parameters. When $\sin(\phi) = \cos(\phi)$, with $\phi = \pi/4$, one process has no or full knowledge about one parameter. With $\phi$ approaching 0 or $\pi/2$, where $\sin(\phi)$ and $\cos(\phi)$ become different, this knowledge increases or decreases, respectively.

## 6 Golden ratio

The solution to the inner Bayes equation is connected to the golden ratio (Livio, 2002), which becomes evident from the transformations of equations above and the assumptions made for both processes. Based on their right-hand equations, both dual processes must meet the same requirement to satisfy the inner Bayes equation, assuming that $\log_\lambda(x)$ produces $x$. For Eq. 12, with $P(B) = P(B|A)$, and for the corresponding Eq. 19 of the dual process, with $P(A) = P(A|B)$, this requirement can be written as

$$p = \frac{1-p}{p}, \tag{24}$$

where the variable $p$ is a placeholder for one of the probabilities. Eq. 24 holds true if $p$ is the golden ratio, which is defined by the equivalent quadratic equation,

$$p^2 + p - 1 = 0, \tag{25}$$

which has two irrational solutions $p_1$ and $p_2$:

$$p_1 = \frac{\sqrt{5} - 1}{2} \approx 0.618, \tag{26}$$

and

$$p_2 = \frac{-\sqrt{5}-1}{2} \approx -1.618 \tag{27}$$

A key observation is that the complement of both solutions, $1 - p$, equals their square:

$$1 - p = p^2 \tag{28}$$

Alternatively, another quadratic equation that may be more frequently encountered in textbooks can be used to arrive at the golden ratio. This equation is obtained by substituting $-p$ for $p$ in Eq. 25:

$$p^2 - p - 1 = 0 \tag{29}$$

The alternative equation also possesses two irrational solutions, namely the negations of $p_1$ and $p_2$:

$$-p_1 \approx -0.618 \quad \text{and} - p_2 \approx 1.618 \tag{30}$$

For these solutions, the complement $1 - p$ is the negative reciprocal:

$$1 - p = -\frac{1}{p} \tag{31}$$

Computing the complement of the golden ratio allows changing viewpoints and switching between the solutions to the inner and outer Bayes equations. This will become important in the next section for training neural networks.

The golden ratio is sometimes represented by the letter $\varphi$ in the literature. It is often defined as a single value, usually $\varphi \approx 1.618$, and negative values are not considered (Livio, 2002; Huntley, 1970). However, each of the four solutions to the aforementioned quadratic equations will be referred to as the golden ratio in the context of this paper.

# 7 Theoretical implications

Supervised training methods first present a teaching input to a neural network and then try to make the network's output the same as the input by adjusting the network weights. This equalizing of input and output can be related to equalizing multiplicands to satisfy the inner Bayes equation. For example, in Eq. 18, the term $P(B|A)/P(A|B)$ can be considered as input and the term $P(A)$ in the lambda expression as output. The task of the lambda expression is then to make both terms the same to satisfy the inner Bayes equation. Moreover, the lambda expression $\log_\lambda\big(P(A)\big)$ becomes the gradient of a linear function for the outer Bayes equation. These relationships help to determine the optimal learning rate and momentum weight for training based on backpropagation and stochastic gradient descent (SGD).

A training method based on backpropagation estimates the gradient of a loss function with respect to each network weight, where the loss function measures the difference between input and network output. Backpropagation methods try to minimize the loss by following the gradient and updating the network weights accordingly (LeCun et al., 2012). They accomplish this for one network layer at a time, iteratively propagating the gradient back from the output layer to the input layer. To move along the gradient towards the minimum of the loss function, a delta is added to each weight, which often has the following form, including a momentum term:

$$\Delta w_{ij}(t) = -\eta \frac{\partial L}{\partial w_{ij}(t)} + \alpha \cdot \Delta w_{ij}(t-1) \tag{32}$$

In (32), $L$ is the loss function, and $\Delta w_{ij}(t)$ denotes the delta added to each weight $w_{ij}$ between a node $i$ and a node $j$ in the network at training iteration (or time) $t$. The term $\partial L/\partial w_{ij}(t)$ is the partial derivative of the loss function with respect to $w_{ij}$, at time $t$, which is multiplied with the learning rate $\eta$. The sign of $\Delta w_{ij}(t)$ is negative, so the loss function approaches its minimum. In practice, a momentum term describing the weight change at time $t - 1$, $\Delta w_{ij}(t-1)$, is commonly added. This term is typically multiplied by a weighting factor $\alpha$, as seen in (32).

The traditional understanding is that the momentum term improves stochastic gradient descent by dampening oscillations. However, the dual process model offers another explanation for the performance improvement brought about by the momentum term. As of yet, a conclusive theory for the optimal values of the learning rate $\eta$ and the momentum weight $\alpha$ has been lacking. Although

second-order methods (Bengio, 2012; Sutskever et al., 2013; Spall, 2000) as well as adaptive methods (Jacobs, 1988; Kingma and Ba, 2014; Duchi et al., 2011; Tieleman and Hinton, 2012) have been tried with various degrees of success, an ultimate answer has still to be found. Both parameters are usually determined heuristically through empirical experiments or systematic search (Bergstra and Bengio, 2012). Training results can be very sensitive to the value of the learning rate. For example, a small learning rate may result in slow convergence, whereas a larger learning rate may result in the search passing over the minimum loss. Negotiating this delicate trade-off in the regularization of the training process can be time-consuming in practical applications. The literature seems to prefer initial learning rates around 0.01 or smaller for SGD, although reported values differ by several orders of magnitude. For the momentum weight, higher initial values around 0.9 are more common (Li et al., 2020; Krizhevsky et al., 2012; Simonyan and Zisserman, 2014; He et al., 2016).

As shown in the following, the proposed dual process model allows deriving theoretical values for both regularization parameters: learning rate $\eta$ and momentum weight $\alpha$. In the weight adjustment given by Eq. 32, each summand represents a gradient of one of the two dual processes. These are the partial derivative $\partial L / \partial w_{ij}(t)$ and the momentum term $\Delta w_{ij}(t-1)$. The momentum weight $\alpha$ follows from the results above, where the lambda expression can be considered as the gradient of the current iteration at time $t$. The other multiplicand of the inner Bayes equation corresponds to the gradient of the other dual process at time $t - 1$, assuming that both dual processes are interleaved, if not in parallel.

The previous sections showed that the inner Bayes equation is met when both summands are equal to $sin(\pi/4) = \cos(\pi/4) = 1/\sqrt{2}$ and when they are equal to the golden ratio. Therefore, the delta at $t - 1$, $\Delta w_{ij}(t-1)$, needs to be multiplied by a constant to obtain the golden ratio. This constant is the momentum weight $\alpha$, which needs to satisfy $\alpha/\sqrt{2} = p_1$, and can thus be computed as follows.

$$\alpha = \sqrt{2} \cdot p_1 \approx 0.874, \tag{33}$$

where $p_1$ is the value of the golden ratio in Eq. 26. So, this logic provides the value of the first regularization term, namely the momentum weight $\alpha$, with $\alpha \approx 0.874$.

The learning rate $\eta$ can be derived from the momentum weight $\alpha$ by converting the latter to the corresponding value for the dual process. The dual process does not aim to satisfy the inner Bayes equation with $\phi = \pi/4$. Instead, it aims to satisfy the outer Bayes equation, with $\phi = 0$ or $\phi = \pi/2$, and thus $\sin(\phi) = 0$ and $\cos(\phi) = 1$, or $\sin(\phi) = 1$ and $\cos(\phi) = 0$. By moving in the opposite direction of the gradient of its dual counterpart, the first process can minimize its loss in satisfying the inner Bayes equation. Accordingly, taking the complement of the momentum weight $\alpha$ twice results in the learning rate $\eta$ for the gradient change at time $t$. Taking the complement of $\alpha$ twice can be understood as looking at the same process from a dual point of view. Mathematically, this can be achieved by squaring the simple complement, $1 - \alpha$. Squaring the complement follows the functional equation of the golden ratio described by Eq.28. Squaring also means bringing the multiplier 2 back in, which was extracted from the lambda expression in Eq. 15 and Eq. 22 to represent all solutions graphically. Applying these steps to the momentum weight $\alpha$ then results in the following equation for the learning rate $\eta$:

$$\eta = (1 - \alpha)^2 \approx 0.016 \tag{34}$$

So, this computation provides the value for the second regularization term, learning rate $\eta$, with $\eta \approx 0.016$.

# 8 Discussion

Starting from Bayes' theorem, this paper develops a theoretical framework that describes any decision of a machine classifier as the result of two processes. The first decision process determines the input message; specifically, it decides whether the input is encoded according to its true value or needs to be inverted. On the other hand, the second decision process decides whether the output should be equal to the input or needs to be inverted. Although both decision processes run simultaneously, they are independent processes, with each possessing knowledge not accessible to the other process. What is uncertain for one process is certain for the other, and vice versa. The first process does not know whether the input should be equal to the output, and conversely, the second process does not know whether the input needs to be inverted. This means a binary decision always involves two bits, one indicating the encoding of the input and the other defining the relationship between input and output.

However, practically, only one of the two processes can be performed at a time, leaving one bit of uncertainty for one of the processes.

Theoretically, the framework proposed here formulates this duality with two processes having different perceptions of zero and one (black and white). The output of one process is the input to the other process. While one process tries to make its output equal to its input, the other aims for the opposite and tries to make its output as different as possible. The mathematical definitions of these processes are defined by the outer and inner Bayes equation, the latter of which is an entangled version of the original Bayes' theorem. By introducing the logarithm, each process is given a control parameter, namely the base of the logarithm, to achieve its goal. This parameter, which is essentially a multiplier, allows each process to control the magnitude of the input/output.

The solution space of the proposed double-Bayesian decision framework can be visualized with the trigonometric functions sin and cos. Furthermore, the golden ratio defines solutions to the inner Bayes equation. Connecting these two observations leads to specific values for momentum weight and learning rate for stochastic gradient descent, which tries to minimize the difference between training input and output during training.

The supplemental material to this paper contains experiments for the MNIST dataset (LeCun et al., accessed May 21, 2024), where the proposed double-Bayesian learning framework is practically evaluated. The theoretical parameters found in this paper did, in fact, provide the best performance for a network trained with stochastic gradient descent in a large grid search for learning rate and momentum weight.

## 9    Conclusion

Three primary characteristics define the work presented in this paper: First, a double-Bayesian approach that understands learning as a process involving two Bayesian decisions instead of a single decision, like in contemporary approaches. Second, solving a Bayesian decision problem is equivalent to finding a fixed point for a logarithmic function measuring uncertainty. Third, the golden ratio defines solutions to a Bayesian decision problem. These three characteristics make the proposed approach novel and unique.

The double-Bayesian framework leads to new theoretical results for training neural networks, particularly specific hyperparameter values for backpropagation and gradient descent. These results are in contrast with other gradient descent heuristics in the literature that either use dynamic hyperparameters or second-order methods for adjusting parameters during training. It will be interesting to see how this conceptual difference will be resolved in the future. The proposed framework offers new ways to understand how neural networks make decisions and may thus contribute to the interpretability and explainability of neural networks, an actively investigated research area.

The proposed framework may also help build bridges to other disciplines like neuroscience or physics. For example, representing all possible solutions to a double-Bayesian decision by means of trigonometric functions, as done in this paper, introduces waves. Incorporating brain waves into machine learning, a feature that traditional machine learning approaches are arguably lacking, would likely entail a better understanding of learning in general. This better understanding could mean training methods for smaller networks that could achieve the same performance with less training data, as motivated at the beginning of this paper.

Another example of a discipline that could be related to this work is quantum mechanics. One of the fundamental concepts in quantum mechanics is Heisenberg's uncertainty principle, which states that certain pairs of physical properties, such as the position and momentum of an electron, cannot be measured with absolute certainty. The more accurately one property is measured, the less is known about the other property. The proposed double-Bayesian framework incorporates such an intrinsic uncertainty and makes a connection to Bayesian decision theory, which could lead to new insights.

Although empirical evidence in the literature supports the theoretical hyperparameter values derived in this paper, and the experiments in the supplemental material show that these values outperform other value pairs, more practical experiments are needed to corroborate these values. To address this limitation, future work will validate the practicality of the derived hyperparameter values in additional experiments across different domains and compare their performance with the performance of other values and other optimization strategies.

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

# A    Appendix / supplemental material

Two grid searches for the publicly available MNIST dataset were performed to corroborate the learning rate and momentum weight derived in the main paper (LeCun et al., accessed May 21, 2024). The MNIST dataset contains gray-scale images of handwritten digits and is one of the prominent datasets used to evaluate machine learning methods. It is split into a training and a test set, where the latter serves as a standard of comparison. Figure 2 shows an example of the MNIST data.

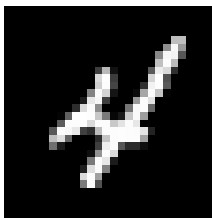

Figure 2: A slightly enlarged example from the MNIST dataset showing a handwritten digit (4).

## A.1    Experiments

The grid searches were performed on the full-size MNIST dataset and a smaller version of MNIST containing only 50% of the training data. In the latter case, a stratified sampling method named *StratifiedShuffleSplit* was used to create a stratified random subset of the training samples (Scikit-learn developers , BSD License; Pedregosa et al., 2011; Buitinck et al., 2013). This ensured that the class distribution in the training subset was the same as in the original full-size training set. The degradation in dataset size allowed observing how each optimizer performed under varying amounts of training data, assuming that providing less training data posed a harder problem.

A deep learning model was trained based on a convolutional neural network (CNN). The model consisted of two convolutional layers, each followed by a ReLU activation function and a max pooling operation. The first convolutional layer had a single-channel input (grayscale image) and applied 16 filters, followed by a second convolutional layer that expanded the channel size to 32. Both convolutional layers used a 3x3 kernel size, a stride of one, and a padding of one. After each convolution, a ReLU activation function introduced non-linearity, and a max pooling operation with

a 2x2 kernel and stride reduced the spatial dimensions by half. A dropout layer with a rate of 0.25 was applied after flattening the output to prevent overfitting. The network concluded with two fully connected layers with a final output of 10 classes, where the maximum output value determined the class of an input image. The number of parameters was around two hundred thousand for an MNIST input image of size 28x28. A weight initialization was performed using the Kaiming uniform method. No data augmentation techniques were applied; however, the input was normalized to the range [-1,1]. The training used a batch size of 64 and was conducted over 30 epochs, employing cross entropy as the loss function. The sizes of the training, validation, and test datasets were 54,000, 6,000, and 10,000, respectively. Finally, the model's performance was assessed through 10-fold cross-validation.

## A.2 Results

The results of both grid searches are shown in Figure 3 for the full-size training set and in Figure 3 for the smaller training set with 50% of the size. The following values were used as momentum

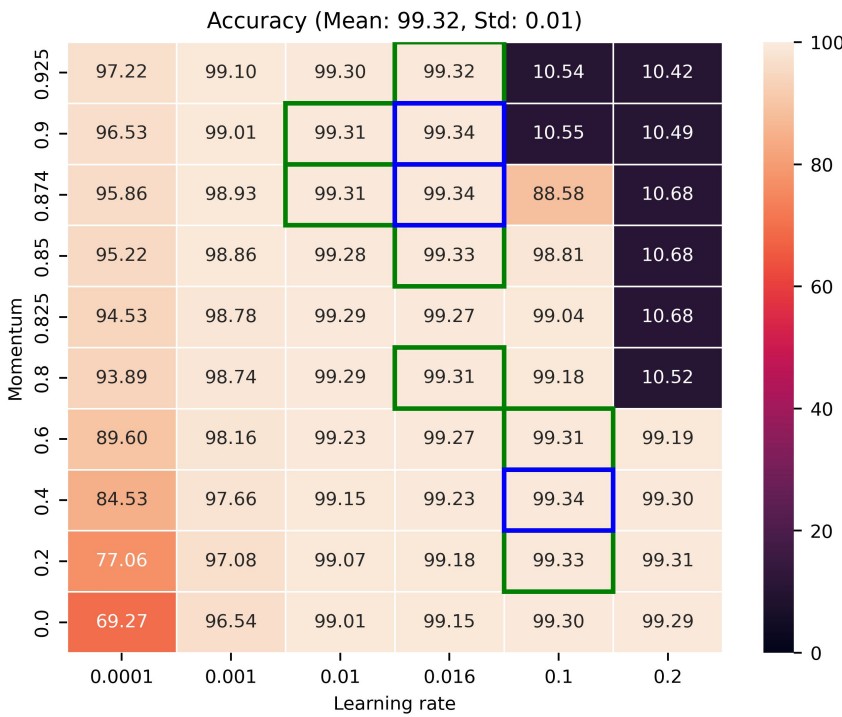

Figure 3: Grid search results for MNIST

weights for each grid search: 0, 0.2, 0.4, 0.6, 0.8, 0.825, 0.85, 0.874, 0.9, and 0.925. On the other hand, the following values were used as learning rates: 0.0001, 0.001, 0.01, 0.016, 0.1, 0.2. These values included the momentum weight derived in the paper ($\alpha \approx 0.874$) and the derived learning rate ($\eta \approx 0.016$). Other values were chosen based on their use in the literature or to increase the resolution around the derived theoretical values. All possible combinations of values span a 6x10 grid. The color of each square in the grids of Figure 3 and Figure 4 represent the performance of the corresponding pair of momentum weight and learning rate, with lighter colors representing higher performance. Green rectangles indicate the top ten performing pairs, whereas blue rectangles show the best-performing pair. Note that more than one pair can share the best performance, as in Figure 3.

Figure 3 shows that no pair of momentum weight and learning rate provides better performance on the full-size MNIST set than the pair derived in the paper, $(0.016, 0.874)$, although this pair has to share its first place with other pairs. The classification accuracies for the reduced training set size are slightly lower in the table of Figure 4, as one would expect for a problem with less training data.

Nevertheless, the theoretical values derived in the paper for momentum weight and learning rate show

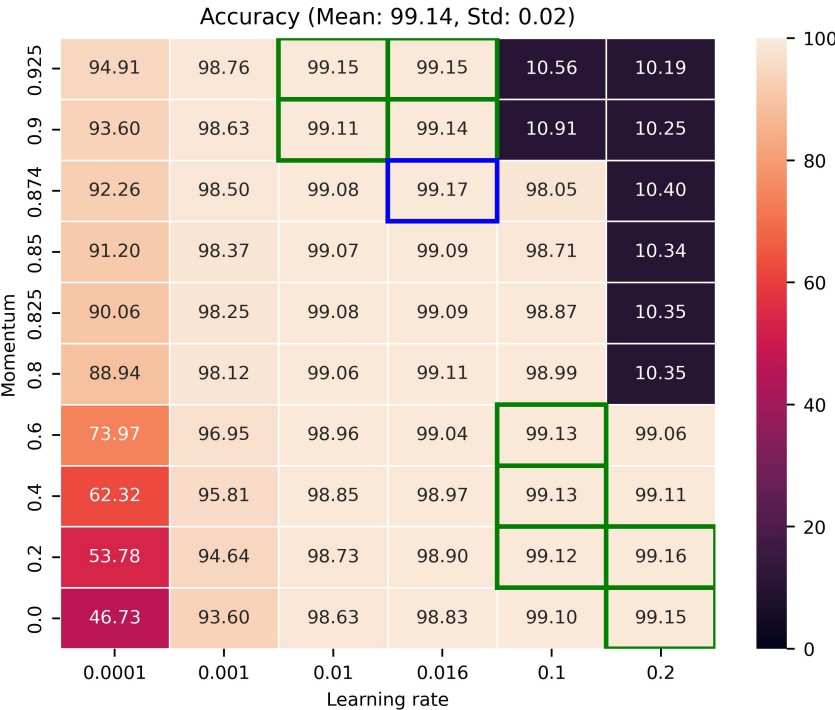

Figure 4: Grid search results for MNIST using only 50% of the training data

800
801 again the best performance.

## A.3  Computational environment and runtime

803 The software was developed using Python 3.10, and the Convolutional Neural Network (CNN) model
804 was implemented in Pytorch 2.2.2. For each combination of learning rate and momentum weight (60
805 combinations in total), the training time was approximately three hours for 100% of the training set
806 size and about 1.5 hours for 50% of the training set. Consequently, the cumulative GPU time for all
807 experiments was approximately $(3 + 1.5) \times 60$ hours, which is 270 hours. The average memory usage
808 was roughly 1 GB for each combination. For more information about the software requirements and
809 workflow, see the Readme file uploaded as supplemental material together with the code.

## A.4  Computing cluster

811 Figure 5 shows an overview of the GPU computing cluster that was available for the experiments,
812 including the type of GPUs among which the processing was distributed.

| GPU nodes | Processor cores per node | Memory | Network |
|:---:|:---:|:---:|:---:|
| 36 | 32 x 2.8 GHz (AMD Epyc 7543p) hyperthreading enabled 256 MB level 3 cache **4 x NVIDIA A100 GPUs** (80 GB VRAM, 6912 cores, 432 Tensor cores) NVLINK | 256 GB | 200 Gb/s HDR Infiniband (1:1) |
| 56 | 36 x 2.3 GHz (Intel Gold 6140) hyperthreading enabled 25 MB secondary cache **4 x NVIDIA V100-SXM2 GPUs** (32 GB VRAM, 5120 cores, 640 Tensor cores) NVLINK | 384 GB | 200 Gb/s HDR Infiniband (1:1) |
| 8 | 28 x 2.4 GHz (Intel E5-2680v4) hyperthreading enabled 35 MB secondary cache **4 x NVIDIA V100 GPUs** (16 GB VRAM, 5120 cores, 640 Tensor cores) | 128 GB | 56 Gb/s FDR Infiniband (1.11:1) |
| 48 | 28 x 2.4 GHz (Intel E5-2680v4) hyperthreading enabled 35 MB secondary cache **4 x NVIDIA P100 GPUs** (16 GB VRAM, 3584 cores) | 128 GB | 56 Gb/s FDR Infiniband (1.11:1) |
| 72 | 28 x 2.4 GHz (Intel E5-2680v4) hyperthreading enabled 35 MB secondary cache **2 x NVIDIA K80 GPUs** with 2 x GK210 GPUs each (24 GB VRAM, 4992 cores) | 256 GB | 56 Gb/s FDR Infiniband (1.11:1) |

Figure 5: GPU computing cluster

