# OpenReview forum: "Double-Bayesian Learning"
_NeurIPS.cc/2024/Conference — Submitted to NeurIPS 2024_

### Official Review · Reviewer_cT9s · 2024-07-01

**Soundness:** 1
**Presentation:** 1
**Contribution:** 1
**Rating:** 1
**Confidence:** 2

**Summary:**

This paper appears to suggest that any decision is composed of two Bayesian decisions and it trys to evaluate the implications of this idea.

I am very confused by this paper and really don't know what to make out of it. For example, the conclusion seems to be only a brainstorming session of random ideas and the rest of the paper does not appear to be much better.

At the very least, it is not well written, at worst the proposed approach does not make any sense.

**Strengths:**

Given that I don't properly understand what exactly the authors want to achieve, I am unable to formulate the strengths of this paper.

**Weaknesses:**

The presentation is very messy. The paper jumps from topic to topic without me understanding their relations to each other.

**Questions:**

see above

**Limitations:**

see above

---

### Official Review · Reviewer_eiHq · 2024-07-09

**Soundness:** 2
**Presentation:** 1
**Contribution:** 2
**Rating:** 2
**Confidence:** 4

**Summary:**

The paper discusses the implications of Bayes' theorem, making assumptions inspired by a thought experiment of communicating a message. Prior (and model) elicitation by solving a fixed point equation is discussed.

**Strengths:**

* The paper takes a fresh look at decision marking under uncertainty, which is at the center of machine learning.
* The generality of the setting makes the discussion applicable to virtually all of ML.

**Weaknesses:**

While I am sensible to the topic of prior and model elicitation from coherence arguments, I believe the paper needs a thorough revision focussing on clarity. While I have some intuition now, it is still not crystal clear to me what the exact goal or claims of the paper are. See bullets below for constructive comments.

## Major
1. Section 4: what is the probability $P$? What is the underlying space and sigma algebra? What are they supposed to represent?
2. Section 4 introduces several very strong assumptions, like $1-P(A\vert B) = P(B\vert A)$ (is it for all $A,B$ in some sigma-algebra or for a specific pair of events?), that are motivated by an analogy about communicating a message. It is not clear why I should be prepared to make these strong assumptions. The fact that I don't know what $P$ is supposed to model or serve as does not help. Is it a joint probability over the variables describing a decision problem, as in decision theory? In that case, will it be used in conjunction to a loss function to make decisions? Will it be judged by some measure of decision accuracy? Or are we in a de Finetti framework, coming up with a personal probability $P$ which we will use to make predictions about unobserved variables? My intuition is that we are dealing with the latter kind, but this should be explained. And the strong assumptions need to be motivated by more than an analogy about communication.
3. The information analogy which motivates imposing the fixed point equation (9) is unclear, as well to what probability and what events it should apply.
4. p5 L179: the sentence about the parameter being a dynamic parameter for a learning system is unclear. We haven't discussed any learning algorithm yet.
5. I am not sure I see where Eqn (11) comes from. $\lambda$ has been chosen to derive (10) from Bayes' theorem, but it doesn't have to be the right base to write (11), right? Same remark for (18).

## Minor
1. p7 L248: Although neural networks have been a popular class of models and algorithms, supervised learning is not synonymous with neural network training.
2. p7 L252: the meaning of "the $\lambda$ expression" is unclear.

**Questions:**

* Can you formally rephrase the goal and claims of the paper?
* Can you explain what $P$ is representing? Is it a personal probability in the spirit of de Finetti, or a model of the data generating process? Or maybe something else?
* Can you formalize and list the assumptions you make on $P$, and justify them in the context of predicting a categorical variable?

**Limitations:**

This is fundamental work that does not have any immediate negative societal impact.

---

> ### Comment · Reviewer_eiHq · 2024-08-10
>
> I don't see a rebuttal in OpenReview. In any case, I believe that my score would have been hard to move at this stage, and that the manuscript needs a thorough revision before being resubmitted.

---

### Official Review · Reviewer_RMti · 2024-07-19

**Soundness:** 2
**Presentation:** 2
**Contribution:** 2
**Rating:** 4
**Confidence:** 4

**Summary:**

The purpose of this paper is to investigate the optimality of a classifier. It is known that the Bayes classifier is optimal, and it is likewise known that an explicit computation of the Bayes classifier is often very challenging if not impossible. This paper offers an analysis of the Bayes classifier as a sequential solution of two problems. An analysis and interpretation of a vase / faces example is presented and some theory is developed to further understand it. The paper concludes with an application.

**Strengths:**

The authors are exploring an idea which is novel, and the whole thinking about Bayes classifiers as comprising two sub-problems seems novel and worth pursuing.

**Weaknesses:**

I did not really understand the discussion with the vase, the sender and receiver. Perhaps the authors should somehow connect the Bayesian ideas to the description of the problem earlier? I think the paper would really benefit from rewriting Section 4 with the vase as a running example, because it is hard to connect the various decisions with the probabilities. Maybe it's worth to add more illustrations / diagrams for this? The authors are presenting novel ideas and it's hard to understand them as they are currently presented.

For the theoretical implications, I think it would be better to illustrate the approach on a simpler model like a linear one.

The paper started by mentioning the Bayes classifier but does not come back to it as an example.

The paper states that the Bayes classifier is broken up into two decisions, but those are just briefly mentioned in the vase / faces example. The authors should carry this thread of reasoning through the whole paper.

**Questions:**

In line 134, you say that "...if the message is known, then whether the foreground needs to be swapped is unknown." But isn't knowing the message "vase" or "faces" enough? How will swapping the background help?

How are the fixpoint solutions connected to the whole vase / faces example?

**Limitations:**

Yes.

---

### Decision · Program_Chairs · 2024-09-25

**Decision:**

Reject

**Comment:**

Reviewers found the core idea of dual Bayesian decision-making intriguing but struggled to follow the paper's explanations, particularly the vase/faces example, which was poorly connected to the main concepts.  Overall, the paper was deemed not ready for acceptance, with reviewers recommending a thorough revision to improve clarity, coherence, and justification of the proposed ideas.